# Crawl Position Depends on Specific Earlier Motor Skills

**DOI:** 10.3390/jcm10235605

**Published:** 2021-11-29

**Authors:** Ewa Gajewska, Jerzy Moczko, Mariusz Naczk, Alicja Naczk, Barbara Steinborn, Anna Winczewska-Wiktor, Paulina Komasińska, Magdalena Sobieska

**Affiliations:** 1Department of Developmental Neurology, Poznan University of Medical Sciences, 60-355 Poznan, Poland; bstein@ump.edu.pl (B.S.); awwiktor@ump.edu.pl (A.W.-W.); komasinska.paulina@spsk2.pl (P.K.); 2Department of Computer Science and Statistics, Poznan University of Medical Sciences, 60-806 Poznan, Poland; jmoczko@ump.edu.pl; 3Institute of Health Sciences, Collegium Medicum, University of Zielona Gora, 65-417 Zielona Gora, Poland; m.naczk@cm.uz.zgora.pl; 4Department of Physical Education and Sport, Faculty of Physical Culture in Gorzow Wielkopolski, University School of Physical Education in Poznan, 66-400 Gorzow Wielkopolski, Poland; a.naczk@awf-gorzow.edu.pl; 5Department of Rehabilitation and Physiotherapy, Poznan University of Medical Sciences, 61-545 Poznan, Poland; msobieska@ump.edu.pl

**Keywords:** infant, children, neurodevelopment, motor performance, qualitative assessment

## Abstract

Early assessment of motor performance should allow not only the detection of disturbances but also create a starting point for the therapy. Unfortunately, a commonly recognised method that should combine these two aspects is still missing. The aim of the study is to analyse the relationship between the qualitative assessment of motor development at the age of 3 months and the acquisition of the crawl position in the 7th month of life. A total of 135 children were enrolled (66 females). The analysis was based on physiotherapeutic and neurological assessment and was performed in the 3rd, 7th and 9th months of life in children, who were classified according to whether they attained the crawl position or not in the 7th month. Children who did not attain the crawl position in the 7th month did not show distal elements of motor performance at the age of 3 months and thus achieved a lower sum in the qualitative assessment. Proper position of the pelvis at the age of 3 months proved to be very important for the achievement of the proper crawl position at the 7th month. Failure to attain the crawl position in the 7th month delays further motor development. The proximal-distal development must be achieved before a child is able to assume the crawl position. Supine position in the 3rd month seemed more strongly related to achieving the crawl position than assessment in the prone position.

## 1. Introduction

Motor development, particularly the analysis of the achievement of subsequent milestones in infants, is seen as one of the most critical indicators of normal general development, and its assessment is used to predict further motor development [1,2,3]. Robson, in his research, demonstrated a relationship between the time of the occurrence of independent walking and the age at which earlier, quantitative motor skills (sitting, crawling, standing) were acquired. He proved that the assessment of the attainment of earlier motor skills could be predictive for determining the age of walking, preceded by the erect position in the 9th month [2].

The attainment of all milestones results from the development of proper posture, neuromuscular control and the achieved experience in a typically developing nervous system. Crawling (on hands and knees) is an example of the significance of neuromuscular and postural control, as stated by Patrick et al. It is at that time that infants use their upper limbs and knees optimally [4].

Publications show that independent crawling may be achieved between the 8th and 9th months [1,5,6,7,8]. However, there are no publications focused on analysing the crawl position as a prelude to crawling. It is known that the crawl position (raising the trunk over the surface, on extended upper limbs with open palms and knees) is achieved in the 7th month of life, and it is preceded by high support on extended upper limbs with open palms and raising the chest high [1,9].

The spine curvatures that will achieve full expression in the upright position can be observed first at three months. At the same time, the pelvis should be in the intermediate position (no anteversion, no retroversion), and the head performs isolated movements. Upper limbs are raised above the substrate to the body centerline and positioned intermediately with an open palm, while lower limbs are bent in the hip and knee joints at the right angle, raised above the substrate, with feet in the intermediate position. The same qualitative pattern is found in the crawl position, when a child must additionally overcome the force of gravity [1,9,10,11] and will, in the future, enable the child to crawl.

Some authors suggest that in the crawl position is a mirror image (in relation to the substrate) of the supine position at the age of 3 months, when the exact spatial positioning of the body parts is considered (Figure 1) [1,9].

### Why Is the Crawl Position So Important?

The achievement of the crawl position in the ontogeny of motor skills makes it possible for a child both to crawl (move forward) and to sit correctly; these two functions are strongly interrelated, sometimes even regarded as one ability [12]. The crawl position is the first stage of the process of walking on all fours as a superior form of locomotion. It makes it possible for a child to acquire new cognitive experiences, which is of great significance for intellectual development [13]. According to Righetti, the crawl position and subsequently crawling on hands and knees is an early pattern of human infant locomotion, which offers an exciting way of studying quadrupedalism in one of its simplest forms [14].

It seems important to underline that crawling is not a final (highest) form of locomotion, just a stage of gaining complete control over alternate bipedal locomotion.

The authors aimed to assess which elements of the motor performance at the age of three months, evaluated with the use of previously described evaluation sheet [10], are related to the motor performance at the age of seven months (summarised as the crawl position) and subsequent achievement of erect posture at the age of nine months.

## 2. Methods

### 2.1. Participants

A general practitioner or a paediatrician referred children to the Clinic of Neurology for a periodic assessment of the development, in some cases also because of parents’ concerns. Children born preterm were assessed at the corrected age [15].

Ultimately, 135 children were enrolled in the study (born at gestational age 38 ± 3; 66 females, 69 males). The exclusion criteria were genetic disorders, metabolic disorders, and severe congenital disabilities. There were 96 children born at term (gestational age 39.5 ± 1.2) and 39 children born preterm (gestational age 34.2 ± 2.7). None of the children under investigation had microcephaly or macrocephaly.

### 2.2. Procedure

The functional assessment was performed by a physiotherapist, who assessed 3-month-old children (at least 12 weeks completed; in the case of preterm babies, corrected age was considered), according to the previously described “Quantitative and qualitative assessment sheet” [10,16,17,18]; only the qualitative assessment results were analysed.

The qualitative assessment in the 3rd month of life included 15 elements in the prone and 15 in the supine position. In the prone position, the assessment involved isolated head rotation, arm in front, forearm in an intermediate position, elbow outside of the line of the shoulder, palm loosely open, thumb outside, spinal cord segmentally in extension, scapula situated in medial position, pelvis in an intermediate position, lower limbs situated loosely on the substrate, foot in an intermediate position. In the supine position, the assessment involved head symmetry, spinal cord in extension, shoulder in a balance between external and internal rotation, wrist in an intermediate position, thumb outside, palm in an intermediate position, pelvis extended, lower limb situated in moderate external rotation and lower limb bent at the right angle at hip and knee joints, foot in intermediate position lifting above the substrate. For symmetrical parts of the body, both sides were assessed to exclude asymmetry.

Each element was assessed as: 0—element performed only partially or entirely incorrectly, or 1—element performed entirely correct. The duration of the examination performed by the physiotherapist was between 10 and 15 min. Each assessed element had to be observed at least three to four times during the test. A maximum of 15 points could be given for the prone position and a maximum of 15 points for the supine position.

Interobserver reliability ranged from 0.870 to 1.000 while intraobserver reliability was equal to 1 [16].

In the 7th month, only reaching the crawl position was assessed (YES—if a child showed it at least three times within 15 min of observation, or NO), and according to the results, the characteristics from the 3rd month of life were checked retrospectively for all children.

Neurological assessment: a neurologist assessed all children at the seventh month, based on the Denver Development Screening Test II (DDST II) [19,20] and the assessment of the reflexes, hypotony/hypertony, and symmetry, as suggested by Touwen [21]. After conducting the examination, neurologists classified a child into one of three groups: typical (no neurological abnormalities), suspected (not requiring rehabilitation for observation) or abnormal. A child was classified as abnormal if he/she exhibited apparent neurological disorders, such as increased (hypertony) or decreased (hypotony) muscle tone accompanied by abnormal reflexes and failure to perform tasks in motor skills for a given age group in the DDST II test. A child was classified into the suspected group not requiring rehabilitation for observation if he/she exhibited mild symptoms of neurological disorders, such as mild muscle tone regulation disorders, slight reflex dysfunction, minor developmental asymmetry and a delay in the area of motor skills in the DDST II test [20,21]. Two neurologists with 20 years of clinical experience took part in the project. The procedure was also used in previous papers [10,16,17,18].

In the 9th month, a neurologist assessed the children again, assigned them to the maximum reached motor performance level (expressed as months), and identified children suspected of cerebral palsy. In their case, the final diagnosis was confirmed at the age of 18 months, according to Reference [22], but their further motor development was not analysed in this paper.

The neurological examination was performed independently of the physiotherapeutic assessment. Both the neurologist and the physiotherapist had information only about whether an infant was born preterm or at term to calculate the corrected age, but they were unaware of the infant’s clinical history details or the parallel opinion.

Previously, this type of examination was used in the assessment of children aged three months and the comparison between physiotherapeutic and neurological assessment showed high agreement, with high conformity coefficients (*z* = −5.72483, *p* < 0.001) [16].

### 2.3. Data Analysis

Demographic data showed normal distribution and thus was described as mean ± SD and analysed with the Welch test due to heterogenic distribution of variance. Data from the assessment sheet (ordinal variables) was described with a median, and quartiles and was analysed with non-parametric tests (Mann–Whitney U-test in the case of two groups or Kruskal–Wallis ANOVA in the case of more than two groups); the differences between nominal variables were analysed with the StatXact program, which computes the exact *p*-values to any desired level of accuracy.

## 3. Results

Demographic data for the whole group, divided according to being born at term or preterm, are shown in Table 1.

The study of the motor performance at the age of three months showed no statistically significant difference related to sex between children born at term and infants born preterm (the preterm ones were assessed at a corrected age).

The analysis first included elements of the qualitative assessment at the age of 3 months, in the prone and supine positions, respectively, in children classified according to whether they attained the crawl position at seven months or not. The results are found in Table 2 for the prone position and in Table 3 for the supine position. Statistical significance showed which elements from the 3rd month of life were most important for attaining the crawl position. For the prone position, the thumb outside position, positioning of the pelvis, the lower limbs on the substrate, and the foot placement proved to be significant. Greater statistical differences were found in the supine position. These were: intermediate position of the wrist and the palm, and the thumb outside. As far as the pelvic girdle and the lower limbs are concerned, the differences relate to the position of the lower limbs on the substrate and raising them above the substrate, with a bend in the hip and knee joints and the intermediate foot position.

In Table 4, children are divided according to the result of neurological assessment that took place when children achieved the 9th month of life. Children who developed properly were assessed as 9th month, and for those who were delayed, the highest achieved level was shown. The next columns show their characteristics in the 7th month of life (functional: crawl position, and neurological), and the last column describes them when they were 3 months old (qualitative functional assessment).

Typical motor performance at the age of three months was noticed in children who showed the crawl position at seven months and achieved the level of 9 months. Those children, who achieved the level of development for the age of 9 months in the neurological assessment but failed to show the crawl position at the age of 7 months, had a lower score at the age of 3 months, and there were only four of them. It is worth noticing that they achieved better scores in the supine position than in the prone position. Children, whom the neurologist assessed as below the level for seven months, failed to achieve the crawl position.

However, it should be noticed that delayed children could also score high in the third month, except for children suspected of cerebral palsy (Cp).

## 4. Discussion

The suggested assessment system allows a more detailed description of particular movement elements, and thus it may serve as a diagnostic test, but also as it indicates the main motor problem, it may be used for therapy planning. There are some other suggestions, e.g., the general movements assessment (GMsA). Observations of the quality of general movements are performed in order to determine the integrity of the central nervous system in infants [23]. However, the GM assessment is difficult to perform under outpatient clinic conditions (it is long-lasting and requires recording and re-analysis), and it cannot be used to plan therapy.

In our previous paper [18], we showed that the pelvis position in the 3rd month seemed crucial for the proper attainment of independent sitting or erect posture. The same importance of the pelvis position was confirmed in this paper. However, what is necessary for the support and thus acquiring the crawl position is assuming the proper position of the distal elements of the upper limbs and entire lower limbs. Acquiring the crawl position at the age of seven months is preceded by high support in the 6th month of life. Thus, the shoulder girdle achieves adequate strength first, which guarantees to lift the chest off the substrate and stable, symmetrical support on both extended upper limbs and extended palms at the same time. This feature is probably related to the fact that the grasp reflex declines in the 6th month. It was already described in the paper [16] that properly extended palms in the 3rd month of life were the most differentiating factor in children who achieved or failed to achieve support at six months. The lower part of the body must adequately overcome the force of gravity between the age of 6 and 7 months: raising the pelvis above the substrate requires increased strength of the pelvic girdle muscles and lower limbs, which still operate at this stage homologically [1,9]. Alternation will occur first at the stage of walking on all fours [24]. Interestingly, the differentiation between the crawl position and crawling itself is found in the Hammersmith test for children who have spinal muscular dystrophy: in the majority, they can attain the crawl position but fail to crawl due to the weakness of the crawl position the muscles. Thus, attainment of the crawl position is due more to the spatial planning of the movement and is genetically encoded in the properly developing central nervous system (in contrast to Cp), whereas crawling itself is the next step, also requiring appropriate muscle strength (lacking in SMA) [25].

In order to achieve the crawl position, not only craniocaudal development has to be completed (which should take place by the 3rd month of life), but also the proximal-distal development, occurring later, must be in place. What is necessary is full maturity of the peripheral elements of the motor system, i.e., proper positioning of the hands (open with the thumb outside) and adequately positioned feet (Table 2 and Table 3).

Four children assessed by the neurologist as demonstrating typical motor development at the age of 9 months failed to achieve the crawl position at the age of 7 months, although these children were assessed by the neurologist as typically developed or requiring observation; their qualitative score at the age of 3 months was low (Table 4). It may be presumed that despite this fact, these children achieved the proper level of motor development as they were not inflicted with severe risk factors (one was born preterm). It should be noted that their qualitative score at the age of 3 months was definitely higher in the supine position and lower in the prone position [10,16,17,18]. Our own therapeutic experience shows that “lazy” development of the crawling function may be due to insufficient motor activity in the prone position, and it undergoes self-improvement if parents make it possible for the child to move on the belly early enough.

Analysis of the crawl position in adults may be a form of spine pain diagnostics when proper spine curvatures are checked (opinion of Janda V et al., cited by Wallden et al. [26]). A similar thesis was presented in one of the previous papers that proper motor development at the age of 3 months may determine, but does not guarantee, proper further development [10] (a condition necessary, but not sufficient).

## 5. Conclusions

Proper position of the pelvis at the age of 3 months (both in the supine and prone positions) proved to be very important for achieving the proper crawl position at the 7th month, showing the importance of the craniocaudal development. Elements such as “thumb outside” and “proper positioning of lower limbs” show that the proximal-distal development must be achieved before a child can assume the crawl position. For the first time in our studies, the supine position in the 3rd month seemed more predictive for the crawl position than assessment in the prone position.

## Figures and Tables

**Figure 1 jcm-10-05605-f001:**
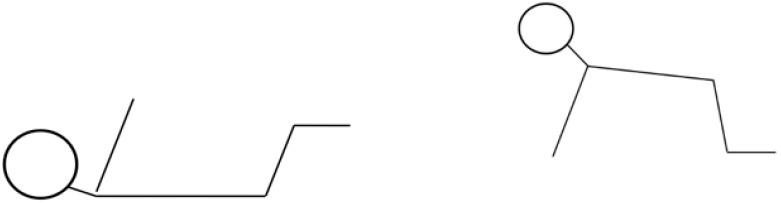
Supine position at the age of 3 months and crawl position at the age of 7 months. Similarities in the spatial positioning of the parts of the body (head, trunk, limbs) can be noticed.

**Table 1 jcm-10-05605-t001:** Demographic characteristics of the studied newborns.

Studied Cases	Term Newborns, *n* = 96	Preterm Newborns, *n* = 39	*p*-Value
Sex: female/male	51/45	18/21	ns
Body weight	3476 ± 468	2238 ±767	*p* = 0.001
Head circumference	34 ± 2	32 ± 3	*p* = 0.001
Body length	55 ± 3	48 ± 5	*p* = 0.001
Apgar 5th minute; good; semi-severe; severe, *n*=	95; 1; 0	32; 6; 1	*p* = 0.001
Born vaginally; Ceasarian section; forceps; vacuum, *n*=	64; 32; 0; 0	17; 20; 1; 1	ns

**Table 2 jcm-10-05605-t002:** The difference between children who achieved (*n* = 92) or not (*n* = 43) in the crawl position in the 7th month in the performance (Y/N) of 15 elements of the qualitative characteristics in the prone position in the 3rd month of life. Exact *p*-value was calculated.

Qualitative Characteristics in the Prone Position	Side of the Body	Crawl Position YES	Crawl Position NO	*p*-Values
Isolated head rotation, Y/N	-	75/17	30/13	0.1815
Arm in front, forearm in the intermediate position, elbow outside of the line of the shoulder, Y/N	Right	67/25	30/13	0.4312
	Left	69/23	31/12	0.3833
Palm loosely open, Y/N	Right	83/9	34/9	0.0691
	Left	83/9	34/9	0.0691
Thumb outside, Y/N	Right	91/1	33/10	<0.001
	Left	89/3	33/10	<0.001
Spinal cord segmentally in extension, Y/N	-	70/22	29/14	0.1606
Scapula situated in the medial position, Y/N	Right	68/24	30/13	0.3797
	Left	71/21	30/13	0.2368
Pelvis in the intermediate position, Y/N	-	85/7	32/11	0.0060
Lower limbs situated loosely on the substrate, Y/N	Right	91/1	35/8	0.004
	Left	91/1	35/8	0.004
Foot in the intermediate position, Y/N	Right	92/0	35/8	0.007
	Left	92/0	35/8	0.007

**Table 3 jcm-10-05605-t003:** The difference between children who achieved (*n* = 92) or not (*n* = 43) in the crawl position in the 7th month in the performance (Y/N) of 15 elements of the qualitative characteristics in the supine position in the 3rd month of life.

Qualitative Characteristics in the Spine Position	Side of the Body	Crawl Position YES	Crawl Position NO	Chi^2^ Values	*p*-Values
Head symmetry Y/N	-	83/9	33/10	3.26	0.3630
Spinal cord in extension, Y/N	-	76/16	31/12	1.31	0.1207
Shoulder in balance between external and internal rotation, Y/N	Right	80/12	32/11	2.34	0.0618
	Left	80/12	32/11	2.34	0.0618
Wrist in the intermediate position, Y/N	Right	89/3	35/8	9.08	0.0017
	Left	89/3	35/8	9.08	0.0017
Thumb outside, Y/N	Right	92/0	35/8	17.21	0.002
	Left	91/1	35/8	13.88	0.001
Palm in the intermediate position, Y/N	Right	88/4	35/8	7.32	0.0042
	Left	88/4	35/8	7.32	0.0042
Pelvis extended (no anteversion, no retroversion), Y/N	-	84/8	31/12	6.97	0.0048
Lower limb situated in moderate external rotation, Y/N	Right	89/3	32/11	13.21	<0.001
	Left	89/3	31/12	15.40	<0.001
Lower limb bent at a right angle at hip and knee joints, foot in intermediate position lifting above the substrate, Y/N	Right	87/5	33/10	7.57	0.0037
	Left	86/6	33/10	6.21	0.0074

**Table 4 jcm-10-05605-t004:** All investigated children were classified according to the maximal developmental level, acc. to neurological investigation in the 9th month. In the second column is the number of children who did or did not achieve the crawl position in the 7th month. In the third column is the number of children classified by the neurologist in the 7th month as normal, suspected or abnormal. In the last column are the median and quartiles of the qualitative examination in the 3rd month, in the prone and supine positions are given. If the number of children was below 7, particular results, not median, are given.

Maximal Level in the 9th Month of Life, Acc. to Neurological Assessment	Functional Assessment: Crawl Position at 7th Month	Neurological Assessment in 7th Month: Normal-Suspected-Abnormal (*n*=)	Sum of Elements in Prone and Supine Positions, 3rd Month; Me (Q25–Q75)
9th month, *n* = 80	NO: *n* = 4	2–2–0	6, 6, 7, 8 7, 11, 12, 15
	YES: *n* = 76	75–1–0	15 (13–15) 15 (15–15)
8th month, *n* = 13			no children
	YES: *n* = 13	10–3–0	12 (9–15) 15 (9–15)
7th month, *n* = 20	NO: *n* = 18	12–4–2	15 (15–15) 15 (15–15)
	YES: *n* = 2	2–0–0	6, 9 11, 13
6th month, *n* = 11	NO: *n* = 11	9–2–0	15 (15–15) 15 (15–15)
			no children
5th month, *n* = 3	NO: *n* = 3	0–3–0	15, 15, 15 15, 15, 15
			no children
Suspected Cp, *n* = 8	NO: *n* = 8	0–0–8	0 (0–0) 0 (0–0)
			no children

## Data Availability

As the data sheet is still under analysis for other publication we decided not to publish it yet.

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
