# Peer review of "Crawl Position Depends on Specific Earlier Motor Skills"

_jcm, 2021, doi:10.3390/jcm10235605_

Round 1

Reviewer 1 Report

Abstract

-Recommend using females and males rather than girls and boys (throughout the paper).

-When sentences begin with a number, the number needs to be written out.

Introduction

-“Proved” is too strong of a word for describing one study. “suggest” or “shows evidence” is preferred

-Additional suggestions for more citations in the paragraph that starts on line 70:

-Kretch, K. S., Franchak, J. M., & Adolph, K. E. (2014). Crawling and walking infants see the world differently. Child Development85(4), 1503-1518.

-Herbert, J., Gross, J., & Hayne, H. (2007). Crawling is associated with more flexible memory retrieval by 9‐month‐old infants. Developmental Science10(2), 183-189.

Methods

-Were there differences between the groups of infants born preterm and to term (in cognitive level, demographics, or age at study)? The results explain that sex was examined, but it would be helpful to look at other variables as well. With the sizable number of those born preterm, seems important to check for general group differences.

-Please add the standard deviation whenever reporting means.

-Line 130 states crawl position was assessed with a yes/no. Could more detail be given on the definition for what signaled “yes” crawl position? Was there a length of time the infant needed to be able to stay in that position?

Results

-Please use <0.001 instead of 0.0000 for small p values

-Table 1- because all motor characteristics are Y/N, seems redundant to have that in each line of the table. A footnote could be added to indicate all qualitative characteristics were measured as yes/no or have the Y/N in the column header for crawl position Yes and Crawl position No. That may also help the reader understand those numbers more readily.

-The paper could be improved by adding a paragraph explaining the results from Table 3. This table was a little confusing and column headers are wordy. For example, it would be more straightforward to have the column header for column 1 be “developmental age”. The table title can tell when the assessment took place.

Discussion

-Could the authors explain why GM assessment is difficult to perform in outpatient conditions?

-Suggest the concluding sentence be organized into one paragraph (lines 265-272).

Reviewer 2 Report

Crawl position depends on specific earlier motor skills by Ewa Gajewska, et al. for Journal of Clinical Medicine

The aim of the study is the assessment of the relationship between the qualitative assessment of motor development at the age of 3 months and the acquisition of the crawl position in the 7th month of life. The main issue is the predictive value of motor skills at 3 months related to the crawl position at 7.

The paper needs of revision.

The Authors wander from descriptive results to predictive results, and this makes not clear the home message of the paper.

Page 3, line from 92 to 100: the data enclosed sound better in a table of “Neonatal data” or characteristics of the studied newborns. The Authors should use mean (SD) accordingly to their statistics, instead of the range here reported, by assuming a normal distribution of the data.

Page 3, line 101: The “majority” should be erased: the number sound better as result. The Authors should include these results in the table of clinical data, similarly the Apgar score.

Page 3, line 111: the Authors summarize the items they studied previously. A table of items and scores seems useful for the reader.

Page 4, It is difficult for audience to take practical home message from tables 1 and 2, a simplification is strongly suggested by including the main meaningful predictive items.

Table 3 is not clear for its aim. Whether it is only descriptive the message of the table is meaningless, otherwise another test should indicate to show the prediction of items in child development.

Page 8, line 258 “According to Janda V. et al…” sounds better. Anyway, it is not cited, and it is not the ref 24.

Page 8, line 263 please the Authors to clarify the sentence into brackets.

 Author Response

Reviewer 3 Report

The article is interesting; still I have to add some comments and recommendations.

  1. Abstract: a short introductory phrase would be welcome.
  2. Introduction: line 71-73 - please reformulate, it is unclear.
  3. Methods: line 92-105 -  consider translating this data into the Results section, and putting it in a table.
  4. Line 197 - what is Cp?
  5. Discussion: Are there other similar studies in the literature? If so, can you make a comparison with them?

Round 2

Reviewer 2 Report

Jcm-1468102

Crawl position depends on specific earlier motor skills-R1

Ewa Gajewska , et al.

Major revision is needed because the paper still requires lot of changes.

Table 1, n of 1st heading is not clear for reader: they are assumed to be “Studied cases (n.)“; second column is: Term newborns n.  or mean (SD); similarly, “Preterm newborns…”; “difference between the subgroups” is redundant in this site, and p value” is enough for the differences; a comparison among preterm and term is debatable for meaning and taken for granted.

Tables 2-4 need to show the same format than table 1.

Tables: “p value” is assumed to be “exact”. Please the Authors to use three digits for decimals.

Table 4: all column titles should be taken easy. “Maximal level…assessment” include methods of evaluation and this is not the right site; similarly for headings in the remaining columns: avoid including the redundant methods, i.e., “Functional assessment…” “Neurological assessment…” or repeat them in the legend

The method/s used to assess the infants’ skills cannot be predictive, because it is new and not standardized. The Authors state “Presented method of assessment is new, prepared basing on practical observation from numerous (are they two??): Vojta and Bobath therapists.

The paper offers some preliminary data. There is a limitation in the reproducibility of this assessment for other infants’ groups and this concern should be more emphasized in the methods and discussion.

Page 8 line 263, “Proper motor development at the age of three months MUST occur if a child is to develop properly further. But some children who showed proper motor development at the age of 3 month will finally show a delay, when assessed at the age of 7 – 8 – 9 months or later, due to various reasons. “

This statement needs some examples or some citation. Otherwise, it seems to counteract the aim of the whole paper. Additionally, it sounds as a meta-communication for reader and requires to be elicited more clearly.

Author Response

Thank you very much for remarks. We made all corrections.

Authors

Reviewer 3 Report

Dear authors,

Congratulations on your work. 

Author Response

Thank you very much for review.
